# Morphological Investigation of Mandibular Lingula: A Literature Review

**DOI:** 10.3390/jpm12061015

**Published:** 2022-06-20

**Authors:** Kun-Jung Hsu, Hui-Na Lee, Chun-Ming Chen

**Affiliations:** 1School of Dentistry, College of Dental Medicine, Kaohsiung Medical University, Kaohsiung 80708, Taiwan; kjhsu1120@gmail.com (K.-J.H.); komschen@yahoo.com.tw (C.-M.C.); 2Department of Dentistry, Kaohsiung Medical University Hospital, Kaohsiung 80756, Taiwan; 3Division of Endodontics and Operative Dentistry, Department of Dentistry, Kaohsiung Medical University Hospital, Kaohsiung 80756, Taiwan; 4Division of Oral and Maxillofacial Surgery, Department of Dentistry, Kaohsiung Medical University Hospital, Kaohsiung 80756, Taiwan

**Keywords:** mandibular lingual, lingula shape, mandibular foramen, ramus

## Abstract

Purpose: The purpose of the study was to review the literature on the shape of the mandibular lingula. Methods: English articles published from 1970 to 2021 in databases (PubMed, Web of Science, and Embase) were selected. Articles meeting the search strategy were evaluated based on the eligibility criteria (participants aged 18 years and over). Dry mandibles and cone beam computed tomography (CBCT) images were used as research materials. The shapes of mandibular lingula were classified as triangular, truncated, nodular, and assimilated. Results: Based on the eligibility criteria, 10 articles (six with dry mandibles and four with CBCT images) were selected for full-text reading and detailed examination. In the dry mandible group, triangular, truncated, nodular, and assimilated lingula were observed on 446, 398, 232, and 69 sides, respectively. In the CBCT group, nodular, truncated, triangular, and assimilated lingula were observed on 892, 517, 267, and 88 sides, respectively. Therefore, the most common lingula types in the dry mandible and CBCT groups were different. The assimilated type was the least common in both groups. Conclusion: In the dry mandible group, the most common lingula type was triangular, followed by truncated, nodular, and assimilated types. In the CBCT group, the most common lingula type was nodular, followed by truncated, triangular, and assimilated types. There were no significant differences in lingula types between the left and right sides of the mandible.

## 1. Introduction

The lingula, located on the mesial side of the ramus of the mandible, is an apophysis at the top front of the mandibular foramen. The bulge or protruding ridge forms a bone spine covering the mandibular foramen, which is referred to as the lingula. The mandibular foramen opens on the mesial surface of the mandibular ramus, which serves as a channel for the inferior alveolar nerve (IAN) and vessels to pass through and enter the mandibular canal [1,2]. Therefore, the position of the lingula is closely related to the positions of the mandibular foramen, inferior alveolar nerve, and vessels within the area. Identifying the actual positions of the lingula and mandibular foramen can help to maximize the chances of success and effectiveness of an IAN block.

The IAN passes through the mandibular body, emerges from the mental foramen, and extends into the lower lip. The IAN provides sensory innervation to the mandibular teeth, lower lip, and the chin surface. When performing ramus surgery, surgeons must be aware of the positions of the lingula and mandibular foramen, avoid the mandibular foramen and mandibular canal, and avoid injury to the IAN or vessels to prevent massive perioperative bleeding or postoperative abnormal sensations and numbness in the lower lip [3]. Sagittal split ramus osteotomy (SSRO) is a versatile technique for ramus procedures. SSRO begins on the inner side of the mandible. The surgeon makes a horizontal cut from above the mandibular foramen, specifically at or above the lingula, splitting the mandible into mesial and lateral bone segments. 

Accordingly, familiarity with the anatomical types of the lingula is essential for understanding IAN block and ramus surgery. By visually inspecting dry mandibles, Tuli et al. [4] classified mandible lingula into triangular, truncated, nodular, and assimilated types according to their shapes and prevalence. Subsequent studies have mostly classified mandibular lingula on the basis of Tuli’s method [4] (Figure 1). This review article explores the classification of adult lingula, the distribution of the types of lingula, and differences among individuals of different sexes and with different skeletal types, as well as the prevalence of bilaterally symmetric shapes.

## 2. Materials and Methods

### 2.1. Search Strategy

We searched the PubMed, Web of Science, and Embase databases for relevant articles. Articles published between 1970 and 2021 were analyzed. The publication language was limited to English. The terms used in our search were “mandibular lingula” and “shape.” The shapes of mandibular lingula (triangular, truncated, nodular, and assimilated types) were based on Tuli’s method [4].

### 2.2. Study Selection and Eligibility

Article selection was performed by two of the authors, who independently read the titles and abstracts and evaluated the articles according to the eligibility criteria. Articles that met the criteria were selected for full-text reading. If the authors disagreed regarding the selection of an article, the entire article was read. The following inclusion criteria were used to screen for articles that: (1) reported randomized control trials, case series, or observational studies and (2) recruited participants aged 18 years and over. Case reports, literature reviews, nonhuman studies, and duplicate articles and studies that involved participants with craniofacial deformity were excluded. Articles that satisfied the aforementioned criteria underwent screening for eligibility. 

### 2.3. Data Extraction and Analysis of Surgical Stability

The aforementioned two authors retrieved the demographic data, methodological data, and stability results from each study independently. If discrepancies were observed in an article, they discussed the article with the other authors.

## 3. Results

### 3.1. Search Strategy

In Figure 2, we searched the aforementioned databases for the term “mandibular lingula” and obtained 322 articles (PubMed, *n* = 126; Web of Science, *n* = 77; Embase, *n* = 119). Thereafter, we searched the databases for the term “mandibular lingula” and “shape” to narrow the scope of search and obtained 61 articles (PubMed, *n* = 22; Web of Science, *n* = 22; Embase, *n* = 17).

### 3.2. Study Selection and Eligibility

Two of the authors read the titles and abstracts of the articles retrieved from the main search independently. After reaching consensus regarding the eligible articles, they manually searched the reference lists of the articles and identified those about adult mandibles involving participants aged 18 years and over. Nonhuman studies and duplicate articles were excluded. Eventually, 51 articles were excluded, and 10 were selected for full-text reading. The data acquired from the selected articles are listed in Table 1 and Table 2. 

### 3.3. Data Extraction and Analysis

Among the studies described in the 10 articles, six used dry mandibles (Table 1) as the research material and four used cone-beam computed tomography (CBCT) images (Table 2). In the dry mandible group, the smallest and largest sample sizes were 50 and 165, respectively, and in the CBCT group, they were 60 and 412, respectively. In the dry mandible group, the minimum and maximum ages of the participants were 18 and 87 years, respectively, and in the CBCT group, they were 18 and 70 years, respectively. Among all the articles (Figure 3), nodular, truncated, triangular, and assimilated lingula were observed on 1124, 915, 713, and 157 sides, respectively. In the dry mandible group, triangular, truncated, nodular, and assimilated lingula were observed on 446, 398, 232, and 69 sides, respectively. In the dry mandible group, the triangular type was the most common lingula type in three of the articles, and the truncated type was the most common in the other three articles. The assimilated type was the least common type in all six of the articles. In the CBCT group, nodular, truncated, triangular, and assimilated lingula were observed on 892, 517, 267, and 88 sides, respectively. In the CBCT group, the nodular type was the most common lingula type in all four of the articles. The assimilated type was the least common in three of the articles, and the truncated type was the least common in one article. In the CBCT group, two articles studied lingula shape in terms of skeletal Class I and Class III. The results indicated that in Classes I and III, the most common lingula type was the nodular type, whereas the least common was the assimilated type.

## 4. Discussion

Tuli et al. [4], through visual inspection, divided 65 dry mandible lingula into four types, namely the triangular, truncated, nodular, and the assimilated types, which have prevalence rates of 68.5%, 15.8%, 10.9%, and 4.8%, respectively. Subsequent studies [6,7,8] have mostly classified mandible lingula according to Tuli’s method. In the dry mandible group, the prevalence of lingula types varied greatly, as reported in previous articles [14,15,16,17]. These discrepancies might be associated with the age, race, dentition, and skeletal type of the samples, and the process of producing and preserving human mandibles. Recent studies [13,14,15,16] have used CBCT images to examine the shape of the human lingula. However, although CBCT images are not affected by the production and preservation processes of dry mandibles, researchers’ interpretations may be affected by the reductions in image sharpness that occur during software processing. Therefore, we analyzed the dry mandible and CBCT groups separately to explore the differences in lingula types among individuals of different sexes, races, and skeletal types, as well as the prevalence of bilateral symmetry. 

In the dry mandible group, the most common lingula type was triangular, followed by truncated, nodular, and assimilated types. The prevalence of the triangular type (446 sides, 39%) was slightly higher than that of the truncated type (398 sides, 34.8%) and the prevalence of the truncated type was higher than that of the nodular type (232 sides, 20.3%). In the CBCT group, the most common lingula type was nodular, followed by truncated, triangular, and assimilated types. The prevalence of the nodular type (892 sides, 50.6%) was significantly higher than that of the truncated type (517 sides, 29.3%), and the prevalence of the truncated type was significantly higher than that of the triangular type (267 sides, 15.1%). Therefore, the most common lingula types in the dry mandible and CBCT groups were different. The only similarity between the groups was that the assimilated type had the lowest prevalence (6% [69/1145] in the dry mandible group and 5% [88/1764] in the CBCT group). 

We compared the prevalence of the lingula types on the left and right sides of the mandible to investigate the prevalence of bilateral symmetry and asymmetry. In both the dry mandible and CBCT groups, the bilateral symmetry was greater than the asymmetry. In the dry mandible group, four articles discussed the prevalence of bilateral symmetry, and symmetrical and asymmetrical samples accounted for 81.1% (642/792) and 18.9% (150/792) of the total samples, respectively. In the CBCT group, two articles discussed the prevalence of bilateral symmetry, and symmetrical and asymmetrical samples accounted for 78.9% (750/950) and 21.1% (200/950) of the total samples, respectively. Investigating the bilateral symmetry, the triangular type was the most common in the dry mandible group, whereas the nodular type was the most common in the CBCT group. Alves et al. [8] reported no significant differences in lingula types on the left and right sides of the mandible. 

Considering sex differences, the lingula types (dry mandible group) in descending order according to their prevalence among men were as follows: triangular (41%, 316/771), truncated (33.6%, 259/771), nodular (19.7%, 152/771), and assimilated (5.7%, 44/771). The lingula types in descending order according to their prevalence among women were as follows: truncated (37.2%, 139/374), triangular (34.8%, 130/374), nodular (21.4%, 80/374), and assimilated (6.6%, 25/374). The most common type of lingula differed between men and women. In three articles in the CBCT group, the lingula type with the highest prevalence among men was the nodular type (50.2%, 429/854), followed by the truncated (30%, 256/854), triangular (15%, 128/854), and assimilated (4.8%, 41/854) types, whereas the lingula type with the highest prevalence among women was the nodular type (52%, 409/786), followed by the truncated (29.4%, 231/786), triangular (14.6%, 115/786), and assimilated (4%, 31/786) types. The most common type of lingula among men and women was the same. Asdullah et al. [9] (dry mandible group) observed no significant differences in lingula types between men and women. However, Alves et al. [8] (dry mandible group) found significant differences in lingual types between men and women: the truncated type was significantly more common among women than men, whereas the triangular type was significantly more common among men than women. Senel et al. [11] (CBCT group) and Jung et al. [12] (CBCT group), who conducted studies on Turkish and Korean populations, observed no significant differences in the lingula types between men and women. Regarding the skeletal type, Jung et al. [12] reported no significant differences in lingula types between Class I and Class III. Because of a lack of substantial information, the relationship between skeletal and lingula types remains inconclusive. 

Exploring ethnic quality, three articles [4,7,9] (dry mandible group) reported that the triangular type was the most common among both men and women in Indian populations. The truncated type was the most common in men and women in two studies [5,6] involving Thai populations [6] and one study [8] of a Brazilian population. Alves et al. [8] reported that, according to their analysis of samples exhibiting bilateral symmetry, the truncated type was significantly more common among women than among men, regardless of race. They [8] further explored differences in the lingula type between different races and sexes and discovered that the truncated type was significantly more common among white women than men and that the nodular type was significantly more common among white men than women. The triangular type was significantly more common among black men than women. The truncated type was more common among black people than among white people, whereas nodular and assimilated types were more common among white people than black people. The truncated type was significantly more common among black men than white men, whereas the nodular type was significantly more common among white men than black men. No significant differences in the prevalence of any of the four types were observed between black and white women. 

There is no report on how lingula morphology affects clinical procedures of SSRO, or other procedures such as IAN block. From an anatomical observation, the assimilated type has no lingula to prevent injury to the inferior alveolar neurovascular bundle during SSRO or IANB procedures. The studies showed a mean lingula height to be 8 to 9 mm [6,8,10] above the mandibular foramen. Therefore, the tip and the measurement of the horizontal osteotomy distance (from anterior ramus border to lingula tip) are easy to identify with lingula morphology such as the triangular, truncated, and nodular types, for a SSRO operation. The order of occurrence (nodular and triangular types) differed between the dry mandible and CBCT groups, possibly due to the blurring of divisions between the nodular and triangular types in CBCT images compared to dry mandible. 

## 5. Conclusions

In the present study, we explored the prevalence of the four major lingula types. First, we classified the articles into the dry mandible and CBCT groups according to the sources of the samples used because the effect of the production and preservation of dry lingula and the clarity of CBCT images after processing might affect researchers’ interpretations. In the dry mandible group, the most common lingula type was triangular, followed by truncated, nodular, and assimilated types. In the CBCT group, the most common lingula type was nodular, followed by truncated, triangular, and assimilated types. No significant differences in lingula types were observed between the left and right sides of the mandible or between people with skeletal Class I or Class III. The observed differences between races and sexes warrant further investigation.

## Figures and Tables

**Figure 1 jpm-12-01015-f001:**
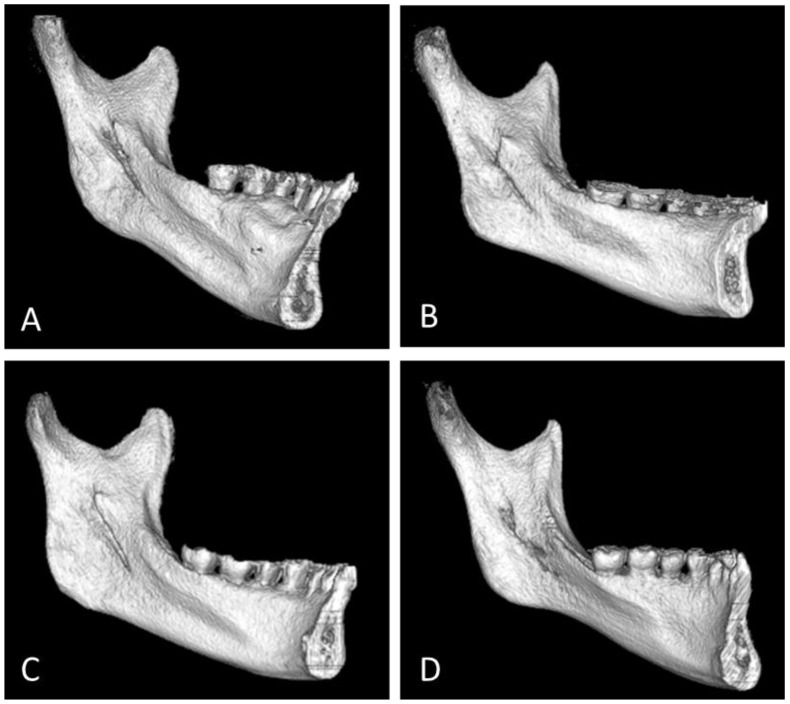
Four types of lingula. (**A**) Nodular (**B**) Truncated (**C**) Triangular (**D**) Assimilated.

**Figure 2 jpm-12-01015-f002:**
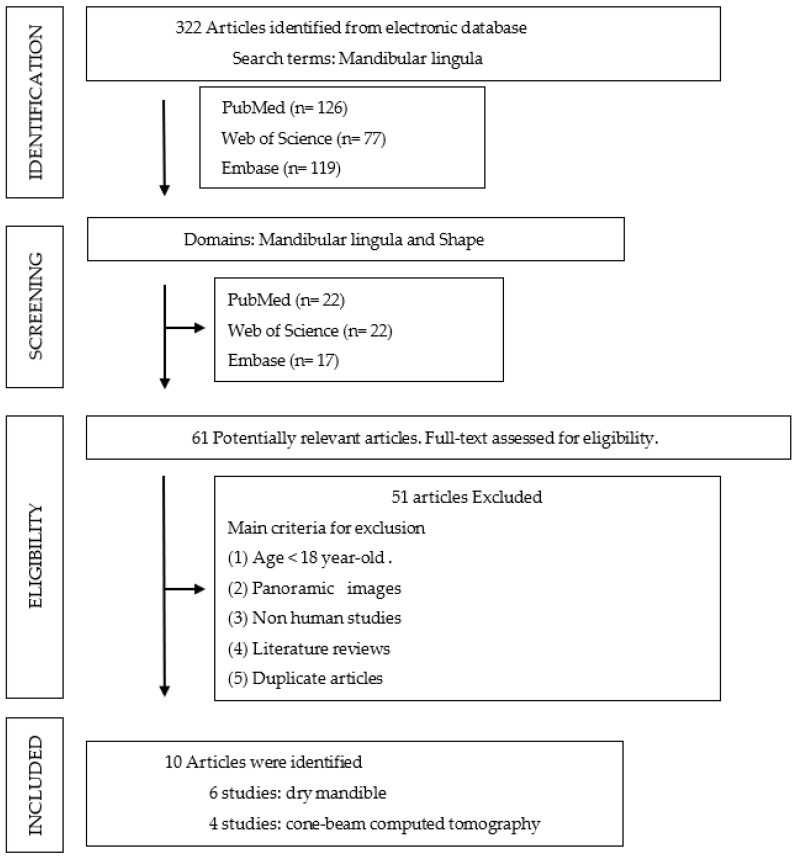
Process flow of article selection in the dry mandible group and CBCT group.

**Figure 3 jpm-12-01015-f003:**
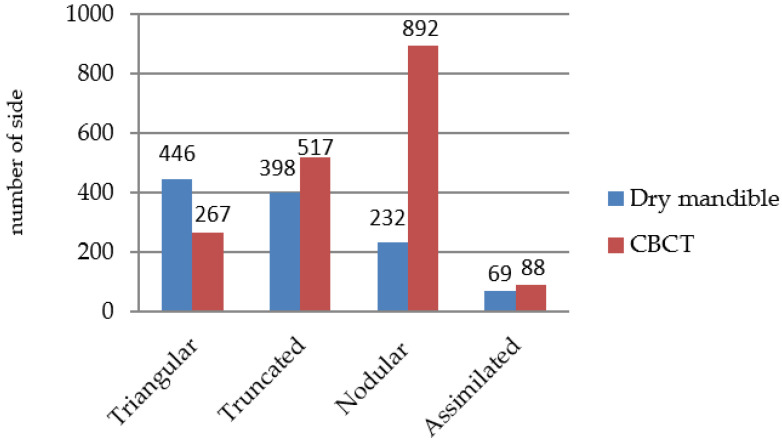
The distribution of lingula types in the dry mandible group and CBCT group.

**Table 1 jpm-12-01015-t001:** Demographic and study characteristics in the included studies (Dry mandible).

Author Year Country of Origin	Material Types	Patients/ Total Sides	Age Mean ± SD (Years)	Sex F (Female) M (Male)	Shapes of Lingula	Number of Sides	Percentage of Total	Number of Bilateral	Number of Unilateral	Number of Unilateral (Right Side)	Number of Unilateral (Left Side)	Male Bilateral (Percentage)	Male Unilateral (Percentage)	Female Bilateral (Percentage)	Female Unilateral (Percentage)
Tuli et al. [4]	Dry	*n* = 165	NA	34 F	Triangular	226	68.5%	220	6	1	5	172 (65.6%)	6 (2.3%)	48 (70.6%)	0
2000	mandible	330 sides		131 M	Truncated	52	15.8%	46	6	5	1	40 (15.3%)	6 (2.3%)	6 (8.8%)	0
India					Nodular	36	10.9%	34	2	0	2	24 (9.2%)	1 (0.4%)	10 (14.7%)	1 (1.5%)
					Assimilated	16	4.8%	14	2	2	0	12 (4.6%)	1 (0.4%)	2 (2.9%)	1 (1.5%)
Kositbowornchai et al. [5]	Dry	*n* = 72	27–87 years	20 F	Triangular	24	16.66%	18	6	4	2	8 (7.69%)	5 (4.81%)	10 (25%)	1 (2.5%)
2007	mandible	144 sides		52 M	Truncated	68	47.22%	58	10	4	6	48 (46.15%)	6 (5.77%)	10 (25%)	4 (10%)
Thailand					Nodular	33	22.92%	24	9	1	8	16 (15.38%)	5 (4.81%)	8 (20%)	4 (10%)
					Assimilated	19	13.19%	14	5	4	1	12 (11.54%)	4 (3.85%)	2 (5%)	1 (2.5%)
Jansisyanont et al. [6]	Dry	*n* = 92	42.4 ± 15.2	34 F	Triangular	55	29.9%	36	19	NA	NA	20 (17.2%)	13 (11.2%)	16 (23.5%)	6 (8.8%)
2009	mandible	184 sides	18–83 years	58 M	Truncated	85	46.2%	68	17	NA	NA	42 (36.2%)	11 (9.5%)	26 (38.2%)	6 (8.8%)
Thailand					Nodular	36	19.6%	26	10	NA	NA	16 (13.8%)	8 (6.9%)	10 (14.7%)	2 (2.9%)
					Assimilated	8	4.3%	2	6	NA	NA	0 (0%)	6 (5.2%)	2 (2.9%)	0 (0%)
Murlimanju et al. [7]	Dry	*n* = 67	Adult	30 F	Triangular	40	29.9%	28	12	6	6	20 (27.0%)	7 (9.5%)	8 (13.3%)	5 (8.3%)
2012	mandible	134 sides		37 M	Truncated	37	27.6%	22	15	9	6	14 (18.9%)	9 (12.2%)	8 (13.3%)	6 (10%)
India					Nodular	40	29.9%	20	20	5	15	8 (10.8%)	12 (16.2%)	12 (20%)	8 (13.3%)
					Assimilated	17	12.6%	12	5	5	0	2 (2.7%)	2 (2.7%)	10 (16.7%)	3 (5%)
Alves and Deana [8]	Dry	*n* = 132	Adult	F: 88 sides				Male	Female	Black	White	Black M	White M	Black F	White F
2016	mandible	253 sides		Black: 62 sides	Triangular	59	23.3%	28.5%	13.6%	24.0%	22.1%	30.2%	26.0%	14.5%	11.5%
Brazil				White: 26 sides	Truncated	124	49.0%	40.6%	64.8%	53.8%	41.0%	47.9%	30.5%	62.9%	69.2%
				Male: 165 sides	Nodular	67	26.5%	29.7%	20.3%	22.2%	33.8%	21.9%	40.6%	22.6%	15.5%
				Black: 96 sides	Assimilated	3	1.2%	1.2%	1.3%	0.0%	3.1%	0.0%	2.9%	0.0%	3.8%
				White: 69 sides											
Asdullahet al. [9]	Dry	*n* = 50	Adult	25 F	Triangular	42	42.0%	36.0%	48.0%	NA	NA	NA	NA	NA	NA
2018	mandible	100 sides		25 M	Truncated	32	32.0%	32.0%	32.0%	NA	NA	NA	NA	NA	NA
India					Nodular	20	20.0%	26.0%	14.0%	NA	NA	NA	NA	NA	NA
					Assimilated	6	6.0%	6.0%	6.0%	NA	NA	NA	NA	NA	NA

*n*: number of patients; NA: Not available; F: female; M: male.

**Table 2 jpm-12-01015-t002:** Demographic and study characteristics in the included studies (Cone-beam computed tomography: CBCT).

Author Year Country of Origin	Material Types	Patients/ Total Sides	Age Mean ± SD (Years)	Sex F (Female) M (Male)	Shapes of Lingula	Number of Sides	Percentage of Total	Number of Bilateral	Number of Unilateral	Number of Unilateral (Right Side)	Number of Unilateral (Left Side)	Male Bilateral (Percentage)	Male Unilateral (Percentage)	Female Bilateral (Percentage)	Female Unilateral (Percentage)
Sekerci and Sisman [10]	CBCT	*n* = 412	Adult	199 F	Triangular	116	14.1%	78	38	12	26	40 (9.4%)	14 (3.3%)	38 (9.5%)	24 (6.0%)
2014		824 side		213 M	Truncated	264	32.0%	204	60	26	34	104 (25.4%)	30 (7.0%)	96 (24.1%)	30 (7.5%)
Turkey					Nodular	422	51.2%	360	62	39	23	188 (44.1%)	35 (8.2%)	172 (43.2%)	27 (6.8%)
					Assimilated	22	2.7%	12	10	8	2	8 (1.9%)	3 (0.7%)	4 (1.3%)	7 (1.7%)
Senel et al. [11]	CBCT	*n* = 63	46	28 F								Male	Female	Male (%)	Female (%)
2015		126 sides	25–70 years	35 M	Triangular	28	22.2%	20	6	5	1	17	11	24.3%	19.6%
Turkey					Truncated	24	19.0%	18	8	2	6	14	10	20.0%	17.9%
					Nodular	41	32.5%	32	9	5	4	18	23	25.7%	41.1%
					Assimilated	33	26.2%	26	7	3	4	21	12	21.0%	21.4%
Jung et al. [12]	CBCT	*n* = 347	27 ± 7.3	166 F, 181 M				Class I	Class I (%)	Class III	Class III (%)				
2018		694 sides	19–50 years	Class I: 190	Triangular	99	14.3%	44	11.6%	55	17.50%	57	42	17.50%	12.70%
Korea				Class III: 157	Truncated	203	29.3%	108	28.4%	95	30.30%	108	95	29.80%	28.60%
					Nodular	375	54.0%	217	57.1%	158	50.30%	188	187	51.90%	56.30%
					Assimilated	17	2.4%	11	2.9%	6	1.90%	9	8	2.50%	2.40%
Akcay et al. [13]	CBCT	*n* = 60	Class I	14 F, 16 M	Triangular	24	20.0%	9	15.0%	15	25.0%	NA	NA	NA	NA
2019		120 sides	18–37 years		Truncated	26	21.7%	18	30.0%	8	13.3%	NA	NA	NA	NA
Turkey			Class III	16 F, 14 M	Nodular	54	45.0%	24	40.0%	30	50.0%	NA	NA	NA	NA
			18–36 years		Assimilated	16	13.3%	9	15.0%	7	11.7%	NA	NA	NA	NA

*n*: number of patients; NA: Not available; F: female; M: male.

## Data Availability

The data used to support the findings of this study are included within the article. The data used to support the findings of this study are available from the corresponding author upon request.

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
