# Peer review of "Morphological Investigation of Mandibular Lingula: A Literature Review"

_jpm, 2022, doi:10.3390/jpm12061015_

Round 1

Reviewer 1 Report

Well organized and reviewed article.  Especially IAN blockage and SSRO are the operations that faced us with lingula.

Which type of lingula ( Nodular, truncated,etc)  makes the procedure anatomically easy could be discussed in literature. Of course lots of factors affected the operation ( SSRO, IAN blockage) ease, but anaotomical factor is also a reason to perform the operation easy. 

Congrats for your hardworking.

Author Response

Reviewer 1

Comments and Suggestions for Authors

  1. Moderate English changes required

Well organized and reviewed article. Especially IAN blockage and SSRO are the operations that faced us with lingula.

Answer: The text had been corrected by a native English speaker and professional text editing service.

  1. Which type of lingula ( Nodular, truncated,etc) makes the procedure anatomically easy could be discussed in literature. Of course lots of factors affected the operation ( SSRO, IAN blockage) ease, but anaotomical factor is also a reason to perform the operation easy.

Answer: In the last paragraph of Discussion section, we add the following sentences.

“There is no report on how lingula morphology affects clinical procedures of SSRO, or other procedures such as IAN block. From an anatomical observation, the assimilated type has no lingula to protect to avoid injuring the inferior alveolar neurovascular bundle during SSRO or IANB procedures. The studies showed a mean lingula height to be 8 to 9 mm above the mandibular foramen. Therefore, the tip and the measurement of the horizontal osteotomy distance (from anterior ramus border to lingula tip) are easy to identify with lingula morphology such as the triangular, truncated, and nodular types, for SSRO operation.”

Reviewer 2 Report

Good work, english is good and the study is appropriate.

Author Response

Reviewer 2

Comments and Suggestions for Authors

  1. English language and style are fine/minor spell check required

Good work, English is good and the study is appropriate.

Answer: The text had been corrected by a native English speaker and professional text editing service.

Reviewer 3 Report

This is the study to investigate and analyze the shape of mandibular lingula showing 4 different morphology of lingula. 

I want to suggest some points. 

1. I want the authors to describe further how lingula morphology affects clinical procedures of SSRO, or other procedures such as IANB. 

2. the paragraph of 116--123 should be in Discussion but not in Result. 

3. It could be great to describe the reason for the different results between CBCT and Dry Mandible. 

4. Please correct the spell in Fig1.

5. Please change the words, black individuals and white individuals. 

6. Fig2, correct the spells in the box, with identification and eligibility. 

7. Fig2. the box of "51 articles Excluded": Age must be lower and equal to 18 but it shows the opposite. 

8. I recommend additionally using graphs to efficiently show the reader the results because it is too difficult to follow the results in the tables. 

Author Response

Reviewer 3

Comments and Suggestions for Authors

This is the study to investigate and analyze the shape of mandibular lingula showing 4 different morphology of lingula.  I want to suggest some points.

  1. I want the authors to describe further how lingula morphology affects clinical procedures of SSRO, or other procedures such as IANB.

Answer: In the last paragraph of Discussion section, we add the following sentences.

“There is no report on how lingula morphology affects clinical procedures of SSRO, or other procedures such as IAN block. From an anatomical observation, the assimilated type has no lingula to protect to avoid injuring the inferior alveolar neurovascular bundle during SSRO or IANB procedures. The studies showed a mean lingula height to be 8 to 9 mm above the mandibular foramen. Therefore, the tip and the measurement of the horizontal osteotomy distance (from anterior ramus border to lingula tip) are easy to identify with lingula morphology such as the triangular, truncated, and nodular types, for SSRO operation.”

  1. the paragraph of 116--123 should be in Discussion but not in Result.

Answer: The paragraph of 116—123 is deleted.  The Discussion section is revised.

  1. It could be great to describe the reason for the different results between CBCT and Dry Mandible.

Answer: In the last sentence of last paragraph in Discussion section, we add the following sentence.

The presentence order (nodular and triangular types) was contrary between the dry mandible and CBCT groups, possibly due to the blurring of divisions between the nodular and triangular types in CBCT images compared to dry mandible.

  1. Please correct the spell in Fig 1.

Answer:  In Fig 1  “Asimilated” is changed to “Assimilated”

  1. Please change the words, black individuals and white individuals.

Answer: black individuals and white individuals are changed to “black people and white people”

  1. Fig 2, correct the spells in the box, with identification and eligibility.

Answer: The spells are corrected in the box in Figure 2.

  1. Fig2. the box of "51 articles Excluded": Age must be lower and equal to 18 but it shows the opposite.

Answer: In the Box of Figure 2   “Age ≧ 18 year-old” is changed to “Age < 18 year-old” .

  1. I recommend additionally using graphs to efficiently show the reader the results because it is too difficult to follow the results in the tables.

Answer: We add the Figure 3 to present the total results from the dry mandible group and CBCT group. The distribution of lingula types in the dry mandible group and CBCT group.

Round 2

Reviewer 3 Report

I could accept this manuscript when the authors revise a few minor parts. 

line 254: "no lingula to protect to avoid..." --> "no lingula to prevent injury to the inferior alveolar neurovascular bundle..."

line 255: " showed a mean height to be 8 to 9 mm.." --> "showed the mean lingula height to be..." And please put REFERENCE. 

line 259: I can't understand what "The presentence order" means. Please change the word so that the readers can easily understand. 

Author Response

Each point of changes at the revised manuscript are underlined with red color.

Reviewer 3

Comments and Suggestions for Authors

  1. line 254: "no lingula to protect to avoid..." --> "no lingula to prevent injury to the inferior alveolar neurovascular bundle..."

      Answer: In the last paragraph of Discussion section, "no lingula to protect to        avoid..." is changed to "no lingula to prevent injury to the inferior alveolar             neurovascular bundle."

  1. line 255: " showed a mean height to be 8 to 9 mm.." --> "showed the mean lingula height to be..." And please put REFERENCE.

        Answer: In the last paragraph of Discussion section, the References (6,8,10)          are added.

  1. line 259: I can't understand what "The presentence order" means. Please change the word so that the readers can easily understand.

      Answer:  In the last paragraph of Discussion section, "The presentence order"  is changed to “The order of appearance (nodular and triangular types) was contrary between the dry mandible and CBCT groups”.

Sincerely yours,

Chun-Ming Chen, DDS, PhD.

School of Dentistry,  College of Dental Medicine,

Kaohsiung medical university, Kaohsiung Taiwan
